# Understanding the relationship between safety beliefs and knowledge for cognitive enhancers in UK university students

**Ngoc Trai Nguyen, Tim Rakow, Benjamin Gardner, Eleanor J. Dommett** *

Department of Psychology, Institute of Psychiatry, Psychology and Neuroscience, King's College London, London, United Kingdom

* Eleanor.dommett@kcl.ac.uk

## Abstract

### Background

Cognitive enhancers (CE) are prescription drugs taken, either without a prescription or at a dose exceeding that which is prescribed, to improve cognitive functions such as concentration, vigilance or memory. Previous research suggests that users believe the drugs to be safer than non-users and that they have sufficient knowledge to judge safety. However, to date no research has compared the information sources used and safety knowledge of users and non-users.

### Objectives

This study compared users and non-users of CE in terms of i) their sources of knowledge about the safety of CE and ii) the accuracy of their knowledge of possible adverse effects of a typical cognitive enhancer (modafinil); and iii) how the accuracy of knowledge relates to their safety beliefs.

### Methods

Students (N = 148) from King's College London (UK) completed an anonymous online survey assessing safety beliefs, sources of knowledge and knowledge of the safety of modafinil; and indicated whether they used CE, and, if so, which drug(s).

### Results

The belief that the drugs are safe was greater in users than non-users. However, both groups used comparable information sources and have similar, relatively poor drug safety knowledge. Furthermore, despite users more strongly believing in the safety of CE there was no relationship between their beliefs and knowledge, in contrast to non-users who did show correlations between beliefs and knowledge.

**Data Availability Statement:** The dataset supporting this research is openly available from the King's College London research data repository at http://doi.org/doi:10.18742/RDM01-690.

**Funding:** The author(s) received no specific funding for this work.

**Competing interests:** The authors have declared that no competing interests exist.

## Conclusion

These data suggest that the differences in safety beliefs about CE between users and non-users do not stem from use of different information sources or more accurate safety knowledge.

## Introduction

Cognitive enhancers (CE), commonly referred to as smart drugs, are prescription drugs taken by individuals, either without a prescription or at a dose exceeding that which is prescribed, to improve cognitive functions such as concentration, vigilance or memory [1]. CE were originally developed to treat a range of disorders including Attention Deficit Hyperactivity Disorder (ADHD), Alzheimer's disease and narcolepsy [2], by targeting various deficits in cognitive functioning such as attention, aberrant learning, and absence of top-down cognitive control [3]. However, they are increasingly being used by healthy individuals to enhance cognition, even though questions remain around their ability to do so within non-clinical populations [2, 4, 5]. One population for whom use of CE is thought to be particularly prevalent is university students, who seek to enhance functioning to improve academic performance [2, 6–8]. The reported prevalence of CE use among students varies markedly, ranging from 5–35% in US studies [8], 1–16% in Continental Europe [9–11], and 9–17% in studies of UK students [12–14].

Several studies have examined ethical, legal and social issues surrounding CE and identified concerns regarding coercion, drug abuse, morality, illegality, fairness and equality [15, 16]. Studies looking at attitudes towards CE indicate that attitudes are impacted by views on these issues, as well as other factors such as stress, learning approaches, competitiveness and awareness of the possibility of using certain drugs to enhance cognition [3, 17, 18]. Safety concerns have also been raised [19]. Perhaps unsurprisingly, studies have found users to be less concerned about the safety of CE than non-users [4, 20], and perceptions of the severity of possible health risks have been inversely associated with willingness to use CE [21–27]. We recently demonstrated that perceived harmlessness and the belief that an individual knows enough about CE to use them safely were significant positive predictors of attitudes towards CE, which in turn predicted CE use in UK university students [12]. However, to date, no study has examined where users and non-users obtain safety information, or the accuracy of their knowledge of drug safety. Furthermore, the relationship between *beliefs* about the safety of CE and *actual knowledge* of drug safety remains unclear.

The studies conducted to date indicate the value of further investigating beliefs and knowledge about safety in key populations. When making any health-related decision, including whether or not to use CE, we might assume that individuals will conduct a cost-benefit analysis of a specific course of action. However, a large body of evidence from decision psychology suggests that, often, people do not take this consequentialist approach in their decisions [28]. Rather, they rely heavily on emotional responses to stimuli and events, in which negative emotions such as fears anxieties and worries are particularly important. Consequently, the features of a decision that elicit these emotions (e.g., risks, and how they are presented) are particularly influential. A related proposal is that negative stimuli may be more salient than positive ones creating a general negativity bias within our decision-making [29, 30]. This negativity bias is supported by a range of evidence, including eye-tracking during reading of risk and benefit health information [31], and may explain why interventions that successfully change risk

perceptions are more likely to result health behaviour changes in contrast to those which focus on benefits. In keeping with these general features of people's decision making, most studies looking at CE tend to focus on perceptions of risk and safety rather than knowledge or evaluations of benefits. There are likely to be several specific reasons for this. Firstly, the perceived safety of drugs can be defined as the absence of risk awareness or risk knowledge [32]. Secondly, there is evidence that the benefits of CE are doubted and perceived as highly variable, even in the student populations where prevalence is high [33], meaning the benefits may be even less informative in this case. Thirdly, when off-label use is considered, including that of the cognitive enhancer modafinil, key professionals (e.g. physicians, regulators) focus on risks not benefits [34, 35].

Based on the evidence that safety is a key issue in research into CE and the prominence of risk over benefits in both theoretical models and health interventions, the present study addresses three novel research questions: comparing users and non-users of CE in terms of i) their sources of knowledge about the safety of CE and ii) the accuracy of their knowledge of possible adverse effects of a typical CE; and iii) examining how the accuracy of this knowledge relates to safety beliefs.

## Method

All procedures were approved by the Institutional Research Ethics Committee (HR15/162824) at King's College London. Written consent was obtained via the anonymous online survey prior to access to the survey.

### Participants and procedure

Eligible participants–i.e., full-time students at the host UK university, aged 18 years or over–completed an online survey. The study was advertised via email circulars to all students at the host institution asking for participants to complete a survey about perceptions of safety and risk of CE. The study was conducted via an anonymous, online questionnaire, hosted by Qualtrics. Study adverts featured a URL linking to the study information and a consent form. Consenting participants were granted access to the online questionnaire, which took approximately 30 minutes to complete. Those who completed the questionnaire were offered entry into a £100 (~$130/€113) Amazon.co.uk voucher prize draw. Data were collected between January and May within the academic year.

### Survey measures

Given the novelty of studying these psychological constructs in relation to CE, few measures existed for assessing them. Therefore, unless indicated below, most items described below were designed for this study. A copy of the full survey items can be found in S1 File.

**Sample characteristics.** Participants were asked to state their gender, age, and the type of qualification they were studying for at the time of the survey, selecting from: undergraduate degree (e.g. BSc), taught postgraduate degree (e.g. MSc) and postgraduate research degree (e.g. PhD). They were asked whether they had taken any of the following with the intent of improving their study results during their current qualification: methylphenidate, amphetamine, modafinil, beta-blockers, rivastigmine. Listings included only drug names rather than brand names in line with previous research [3, 12]. These drugs were all chosen based on frequent citation in the literature on CE use [36–38] and recent work showing use of these drugs to enhance cognition in student populations [2]. Participants who indicated they had taken any of these CE were asked to indicate which they took and the frequency with which they used them (from less than one per year to more than once per week).

**Safety beliefs.** Participants rated their agreement with three statements (1 = strongly agree, 7 = strongly disagree) regarding safety: i) It is important to know whether smart drugs are safe to use ('Safety Importance'); ii) I know enough about smart drugs to judge whether they are safe to use ('Sufficient Safety Knowledge') [3, 12]; iii) I think smart drugs are safe to use ('Safe to Use') (3). Each of these safety belief items was reverse scored, such that a high score indicated strong agreement, and entered into analysis as a separate variable.

**Sources of information.** Participants identified the sources they used when considering safety of CE from: personal experience, experiences of peers, information from websites, social media, NICE guidelines, scientific research and other (please specify). For each of these sources, irrespective of whether they used them, they then rated its reliability (1 = extremely unreliable, 7 = extremely reliable).

**Knowledge of drug effects.** Participant knowledge of the effects of modafinil were assessed using three questions that could be answers using the typical information found in a drug safety leaflet accompanying prescription medication. Note that modafinil was selected because our previous research indicated this was the most commonly used of the listed CE at the host institution, therefore meaning that the participants were most likely to be familiar with this drug [12]. First, participants selected 'true' or 'false', for which of five medical conditions (e.g. diabetes, migraines) modafinil should not be taken with ('Not Safe'). Second, participants identified, in the same way, which of nine conditions would require careful monitoring if modafinil is taken (e.g. depression, heart problems) ('Monitor'). Third, participants indicated the frequency with which fifteen known side effects (e.g. headache, chest pain) of modafinil occurred, choosing from three options (1 in 10, 1 in 100, or 1 in 1000 individuals) ('Side Effects').

## Data analysis

Initial analysis was conducted to characterise the sample by calculating percentages for each level of a categorical variable and the mean for age. Chi-square analysis (gender, qualification) or an independent sample t-test (age) was used to establish whether there were differences between users and non-users for these characteristics. Percentage data was calculated for frequency of use. Given that previous studies found beliefs around safety to differ between users and non-users, we also characterised safety beliefs for the whole cohort and compared between the two groups (Safety Importance; Sufficient Safety Knowledge; Safe to Use). Mean (M) and standard deviation (SD) were calculated for all three safety belief measures (importance of knowing about safety; knowing enough to judge safety and perceived safety) and Pearson's correlation coefficient calculated for relationships between them. Finally, binary logistic regression was used to establish whether the safety beliefs predicted CE use.

Aim 1 was to compare users and non-users in terms of their sources of knowledge about the safety of CE. To do so, three analyses were conducted. Firstly, the total number of sources consulted were calculated and this was compared between the two groups using an independent sample t-test. Secondly, for each of the sources, a Chi-square analysis examined whether there were differences between users and non-users. Finally, perceived reliability of the different measures was compared between users and non-users with a mixed-measures ANOVA followed by post-hoc paired sample t-tests. Aim 2 was to compare the accuracy of safety knowledge in users and non-users of CE. To do this, the total number of correct answers was calculated for each of the three safety questions (Not Safe, Monitor, Side Effects) and independent-sample t-tests were then used to establish group differences. For these and other between-subjects tests of mean difference we assume homogeneity of variance in the population. Analyses reported in S2 File show that no conclusions change if this assumption is not

made. Aim 3 was to examine how the accuracy of participants' knowledge related to their safety beliefs. To achieve this, the Pearson's correlation coefficient was calculated for the relationship between each type of accuracy and each safety belief for the entire cohort and then for users and non-users separately.

# Results

## Sample characteristics

Two-hundred and twenty-two participants expressed an interest in the study and 204 gave consent to participate and accessed the anonymous questionnaire, of which 148 (73%) completed it. All 148 were included in subsequent analysis. The majority of those completing the survey were female (78%); which is a slightly higher proportion than for the overall student population at the host institution where 66% of students are female. Most participants were studying for an undergraduate qualification (62%) but taught postgraduate (26%) and postgraduate research (12%) students were also represented. These proportions are representative of the overall student body at the host institution where approximately 60% of students are undergraduates. The mean age of participants was 23.9 years (SD = 4.88) as expected for a student population. Twenty-one percent (N = 30) of participants reported use of CE during their current qualification. Users were significantly more likely to be male ($\chi^2$ (1) = 4.91, $p$ = 0.027; 34.4% of males; 16.5% of females). There was no difference in terms of qualification between users and non-users ($\chi^2$ (2) = 5.53, $p$ = 0.063). There was also no mean difference in the age of users and non-users ($t$(146) = 1.21, $p$ = 0.227, 95% CI -.0.78, 3.29). The most used drug was modafinil and no users reported multiple drug use (Table 1). Frequency of use varied considerably. However, the most common frequency was once per term (30%) followed by once per month (20%). More frequent use was also found in a substantial proportion (once per week 16.7%; more than once per week 16.7%) with fewer participants reporting infrequent use (once per year 3.3%, less than once per year 13.3%).

When considering all participants (i.e. users and non-users together), mean data indicate there was a strong belief that it was important to know about the safety of CE. Scores on the remaining belief statements suggest a wider range of beliefs and less overall agreement. There were significant correlations between two of the three possible pairings of safety beliefs (Table 2), the strongest of these reflecting that those with higher self-rated knowledge were more likely to rate CE as safe.

Logistic regression analysis (Table 3) found that, together, the three safety beliefs, along with gender (included because gender differences were found in the present study), accounted for 36.7% of the variance in CE use ($\chi^2$ (4) = 39.11, $p$<0.001). Within the model, the belief that CE were safe to use was the only significant predictor of use, such that for each additional 1-point increase on the 1–7 scale the odds of being a CE user increased by more than two-and-

**Table 1. Twenty-one percent of the whole sample reported use of CEs.**

| Cognitive Enhancer | Percentage Reporting Use (%) |
|---|---:|
| Modafinil | 70.0 |
| Beta-blockers | 13.3 |
| Amphetamine | 10.0 |
| Methylphenidate | 3.3 |
| Rivastigmine | 3.3 |

The breakdown of use across the five commonly used drugs within this group is shown here.

**Table 2. Correlations between safety knowledge and beliefs (reversed).**

| Variable | Scale range | Mean (SD) | 1 | 2 | 3 | 4 | 5 |
|---|---|---|---|---|---|---|---|
| 1. Safety Importance | 1–7 | 6.59 (0.71) | - | | | | |
| 2. Sufficient Safety Knowledge | 1–7 | 3.36 (1.69) | -.213** | - | - | - | - |
| 3. Safe to Use | 1–7 | 3.97 (1.43) | -.081 | .398** | - | - | - |
| 4. 'Not safe' Accuracy | 0–5 | 3.18 (1.02) | -.117 | .258** | .232** | - | - |
| 5. 'Monitoring' Accuracy | 0–9 | 6.91 (1.31) | .174* | .033 | .144 | .120 | - |
| 6. 'Side Effects' Accuracy | 0–15 | 5.75 (2.46) | -.004 | .015 | -.192* | -.156 | .057 |

N = 148

* $p < 0.05$

** $< 0.01$

a-half times. To further illustrate the substantial size of this effect: of the 50 participants giving a response below the scale midpoint ('1' to '3') for 'safe to use' only 4% used CE, while of the 58 participants giving a response above the scale midpoint ('5' to '7') 43% used CE (with a 63% prevalence of use among those responding '6' or '7').

## Aim 1: Which information sources do users and non-users draw on?

While the mean number of sources used was higher for users (M = 3.43, SD = 1.14) than non-users (M = 2.97, SD = 1.21), this difference was not statistically significant (t(145) = 1.91, $p = 0.058$, 95% CI -0.95, 0.01) although the effect was small-to-medium in size ($d = 0.039$). Thus, we did not detect a difference between users and non-users in the mean number of information sources used. Using G*Power software, we determined that our sample size was sufficient for 0.95 power to detect a mean difference of one additional data source in the population using a two-tailed test (see Supplementary Materials for full details). Therefore, if a difference of that scale or larger were to exist in the population, it is highly unlikely that we would fail to detect a difference.

The percentage of those using the different types of information source is shown in Fig 1, along with ratings of reliability for all sources. Note that 4.4% reported using 'Other' sources but examination of the specified 'other' sources revealed no consistent sources and therefore, given the low proportion of participants using other sources, this category was excluded from further analysis. Chi-square tests revealed that, for most sources, there were no statistically significant differences between users and non-users for the proportion drawing on the source (experience of peers $\chi^2$ (1) = 3.36, $p = 0.067$; websites $\chi^2$ (1) = 2.47, $p = 0.116$; social media

**Table 3. Logistic regression of safety beliefs about CE as predictors of CE use.**

| Variable | $R^2$ | B | S.E | Wald $\chi^2$ | OR | 95% CI |
|---|---|---|---|---|---|---|
| Constant | .367 | -4.401 | 2.624 | 2.813 | .012 | |
| Gender | | -.903 | .524 | 2.970 | .405 | .145–1.132 |
| Safety Importance | | -.219 | .351 | .389 | .803 | .404–1.598 |
| Sufficient Safety Knowledge | | .157 | .165 | .907 | 1.170 | .847–1.616 |
| Safe to Use | | 1.016** | .280 | 13.122 | 2.761 | 1.594–4.784 |

Note that the table presents the total $R^2$ Nagelkerke statistic

** $p < 0.001$.

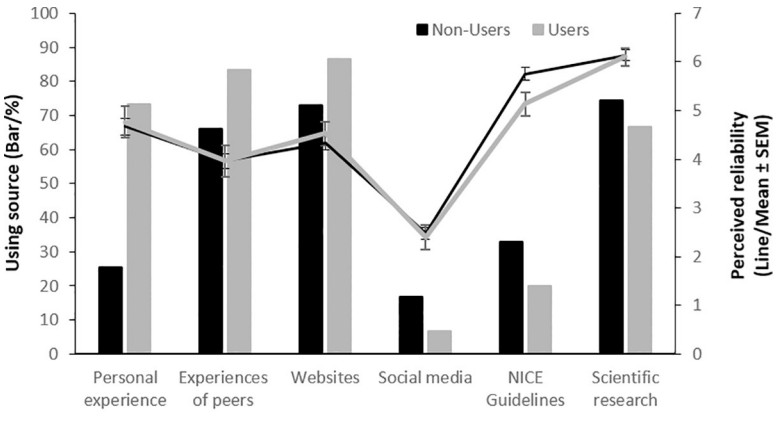

**Fig 1. The frequency of use and perceived reliability for different sources of information about the safety of CE shown by use status.** Bars indicate the percentage using a source whilst lines indicate the perceived reliability of the sources. Note that the most frequently used source (websites) was only the fourth most reliable.

$\chi^2$ (1) = 2.00, $p$ = 0.157; NICE guidelines $\chi^2$ (1) = 1.93, $p$ = 0.165; scientific research $\chi^2$ (1) = 0.79, $p$ = 0.384). The only exception to this was using personal experience ($\chi^2$ (1) = 24.09, $p$<0.001) which was more likely to be used by users than by non-users (73.3% vs. 25.4% respectively). A mixed-measures ANOVA with source as a within-subjects factor, incorporating six levels (i.e. all sources except those labelled as 'Other') and use status as a between-subjects factor was used to examine perceived reliability scores for the different sources. This revealed no significant effect of use status ($F$(1,145) = 0.221, $p$ = 0.639, $\eta^2_p$ = 0.02) but there was a significant main effect of source ($F$(3.41,493.98) = 92.12, $p$<0.001, $\eta^2_p$ = 0.389). There was no use status x source interaction ($F$(3.41,493.98) = 1.16, $p$ = 0.327, $\eta^2_p$ = 0.08. Paired-sample t-tests (corrected $\alpha$ = 0.003) showed there were significant differences between all sources ($p$<0.001) except personal experience and websites ($t$(147) = 1.92, $p$ = 0.057, 95% CI -0.10, 0.65) and experience of peers and website ($t$(146) = 2.84, $p$ = 0.005, 95% CI -0.70, -0.13).

## Accuracy of safety knowledge (Aim 2) and its relation to safety beliefs (Aim 3)

Considering all participants together, accuracy for the 'Side Effects' information was lowest at 38.3% correct (SD = 16.4%), followed by 'Not Safe' information with 63.6% correct (SD = 20.4%). The 'Monitor' information was most accurate with an average of 76.8% correctly identified (SD = 14.5%). For none of these measures did the mean number correct differ significantly between users and non-users (Fig 2): Not Safe, $t$(146) = 1.35, $p$ = 0.179, 95% CI – 0.69, 0.13; Monitor, $t$(146) = 0.72, $p$ = 0.471, 95% CI -0.72, 0.34; Side Effects, $t$(146) = 1.38, $p$ = 0.171, 95% CI -0.30, 1.68. The confidence intervals for these mean differences are for scales with a maximum possible score of 5, 9 and 15, respectively; and therefore the 95% CIs span 19.8%, 11.8% and 13.2% of each scale-range, respectively. We used G*Power software to determine the sample sizes required for 0.8 power to detect an absolute mean difference of 10% of the scale-range for each knowledge scores. Our sample size exceeds the sample size required for this level of statistical power for the monitor and not safe scores, though an additional 56 participants would be required to achieve this high degree of power for the not safe scale (see S2 File).

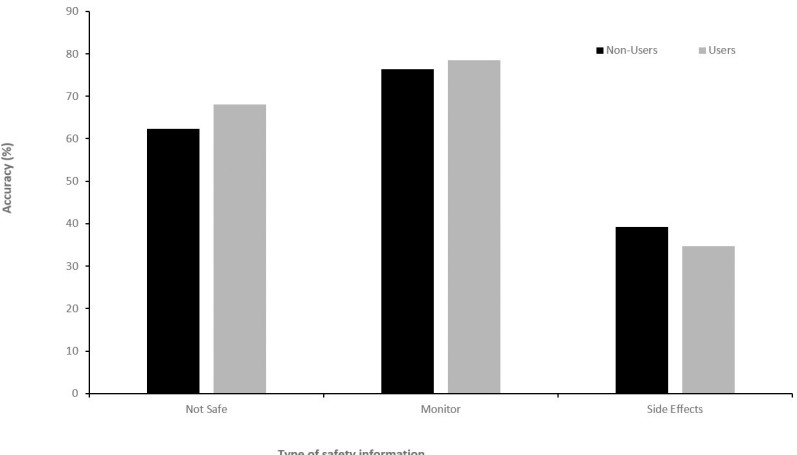

**Fig 2. Accuracy of safety knowledge regarding modafinil was broken down into three categories as is typically found in drug safety leaflets does not differ significantly between users and non-users.**

Pearson's correlation revealed several significant correlations between the accuracy of safety knowledge and safety beliefs when all participants were considered (Table 2). Notably, 'Safe to Use' beliefs positively correlated with 'Not Safe' knowledge but negatively with 'Side Effects' knowledge indicating participants who felt CE were safe to use had less knowledge of side effects. Those with strong 'Sufficient Safety Knowledge' beliefs, i.e. they felt they knew enough to judge safety did have better 'Monitor' knowledge but no other correlations with knowledge were found. Finally, participants with stronger 'Safety Importance' beliefs did have better 'Monitoring' knowledge, but no relationships existed for the other knowledge types. Interestingly, when considering users only, we find that there are no correlations between safety beliefs and accuracy for this group (-.118 < r > .124., all p > .513). By contrast all correlations noted above for the whole group remained when non-users were considered alone except the relationship between 'Safe to Use' and 'Side Effects' which did not reach significant (r = -.169, $p$ = 0.068).

## Discussion

Building on previous research demonstrating that beliefs about safety of CE can predict attitudes, and in turn, use of CE [12], the current study aimed to compare users and non-users of CE in terms of i) their sources of knowledge about the safety of CE and ii) the accuracy of their knowledge of possible adverse effects of a typical CE; and iii) examine how the accuracy of their knowledge relates to their safety beliefs. Before considering the findings in relation to these aims, it is helpful to note that both users and non-users felt that it was similarly important to know about the safety of CE, indicating that the two groups do not differ substantially in the value that they place on safety information. However, users reported stronger agreement that they knew enough about CE to judge safety and (to a substantial degree) that they are safe to use. These findings are in line with previous research [4, 12, 20]. It is typically assumed that the belief that CEs are unsafe prevents individuals taking them, whilst those who perceive them as safe are more likely to take them [10, 22, 23] but the exact relationship between safety beliefs and use is unclear. An alternative explanation for this association between CE use and safety beliefs could arise from the partial dissociation between risk perception and risk taking [39] in combination with the phenomenon of cognitive dissonance [40]. While, in general, we avoid behaviours that we regard as unsafe, this is not always true for all individuals and for all

situations. For example, imagine someone takes CE to get them through an academic 'crisis' even though they have concerns about drug safety. Dissonance theory predicts they would then adjust their beliefs or attitudes to make them more consistent with their recent actions. In this instance, this would mean developing a new belief that CE are not unsafe. Thus, under this account, CE use drives beliefs about safety, rather than the reverse. The reduction of dissonance (discomfort) that is achieved by aligning beliefs with actions may occur without recourse to external sources of information. The nature of the relationship between safety beliefs and CE use therefore still warrants further exploration.

For Aim 1, we found no significant difference in the number of sources used by users and non-users. We note here, however, that our study was not highly powered to detect small differences. Nonetheless, based on the confidence interval reported for this mean difference, it is unlikely that the true difference between users and non-users in the number of information sources is large (e.g., that, in the population of interest, users actually use 1 or 2 more sources on average).The most commonly used sources students used when thinking about safety were scientific research and websites. Interestingly, whilst they rated the former as highly reliable, website reliability ratings were not rated particularly high, ranking only fourth highest out of the six sources for which ratings were given, indicating a mismatch between preferred sources and reliability. For most sources there were no differences between users and non-users in terms of whether the source was used for safety information. The only exception is personal experience which, almost by definition, was higher in users. This additional source may explain the increased belief in safety of CE in users, although this would seem at odds with the reliability of personal experience which was only rated third of six sources. An alternative explanation to source differences underlying different beliefs is that we see motivated reasoning employed when assessing safety information [41], such that users may be more likely to reach the conclusion that the drugs are safe when presented with exactly the same information as non-users. This has been shown to be the case with caffeine, such that coffee drinkers perceive less health risks relative to non-coffee drinkers [42], and therefore may warrant further investigation with regards to CE.

The relatively high use of peer experiences as a source of information may reflect the ready availability and vividness of such anecdotal information [43]. Anecdotes can be an important influence on decisions–including decisions about one's own healthcare when statistical information is also available [44] especially when delivered in a compelling manner. For example, research has shown that evaluative comments about university courses delivered face-to-face by a few (1 to 4) upper-level students had greater influence on students' intentions to enrol in those courses than written summaries of course evaluations from a much larger number of students (26 to 132) who took the course the previous semester [45]. This was also found when the face-to-face comments of (ostensible) students were scripted to match summaries of course evaluations in the written summary reports. Borgida and Nisbett argued that peer experiences are influential because they are vivid (having the character of first-hand experience), and concrete (and therefore prompt action); and because people *intuitively* respond to small samples of information *as if* they were representative of large-sample information i.e., a reliable source [45]. Thus, while it may be unsurprising that our participants cited peer experiences as a common source of information, what is learnt from the current study is that the participants were *explicit* in attributing only modest reliability to this source.

For Aim 2, data showed that knowledge of drug safety was not particularly high with accuracy on questions about conditions that modafinil should not be taken with (63.6%) or would require careful monitoring (38.3%) similar to chance level performance for true/false questions. Students were more accurate at estimating the frequency of known side-effects. Despite users believing that CE are safe to use in comparison to non-users, users were no more (or

less) accurate in their responses to the safety knowledge questions. In terms of the relationship between accuracy of knowledge and safety beliefs i.e. Aim 3, we found that there were no significant relationships between safety beliefs and actual safety knowledge for users, indicating that their belief that drugs are safe to use is not based on more accurate safety knowledge of the drugs. Notably, for non-users there was a significant correlation between their belief about whether the drugs were safe to use and knowledge of which conditions modafinil should not be taken with. Additionally, knowledge of conditions for which careful monitoring was required did correlate with safety beliefs about whether they had enough knowledge to judge safety and whether safety was important. This indicates that for non-users there are some positive correlations between safety beliefs and safety knowledge. This is consistent with previous studies that have shown perceptions of the severity of possible health risks have been inversely associated with willingness to use CE [21–27] and indicates that as well as severity, frequency of side effect may be important.

In addition to providing data to address the specific aims of the study, we report prevalence and frequency of CE use. Here we found that 21% of students surveyed were using CE and using with relatively high *frequency* of use, with the minority of participants being occasional users. The reported prevalence is considerably higher than the 11% [14] and 9.4% [13] reported in previous UK studies, but it is in line with our previous research at this institution [12]. It may be important for the higher rate of use among our participants that the current study was conducted in a more competitive environment than the previous studies, because it has previously been shown, outside of the UK that use of CE is around two times higher at universities with more competitive criteria [46]. Another explanation for the increased prevalence in the current study is that use of CE is increasing, as has been suggested ([13, 47, 48]. The current study showed relatively high *frequency* of use, with the minority of participants being occasional users. The high frequency of use is also somewhat at odds with the low levels of consistent use reported by Singh et al. [13] but is more in line with the media hype around pharmaceutical cognitive enhancement which emphasizes *sustained* use of CE is highly prevalent among UK students [49].

## Limitations

There are several limitations to this study that should be acknowledged. Firstly, the study relied on self-report and therefore, only captures information that participants were willing to report, which may not represent true beliefs and behaviour when relating to drug use [50]. However, it is suggested that self-report can be reliable provided that the information is known to respondents and that the questions are unambiguous, relate to recent activity, require a serious and thoughtful response, and will not lead to embarrassing or threatening disclosures [51, 52]. We believe these conditions were met in the current study because, although drug use could represent an embarrassing or threatening disclosure, the anonymity would have reduced this. Secondly, whilst the overall sample size was sufficient for statistical analysis, the study used a sample from one institution and students self-selected to participate meaning that the results may not generalise to other populations. Thirdly, although our sample size allowed for very good statistical power to detect large effects, the study was not well powered to detect small ones (See S2 File). We may therefore have missed detecting differences between CE users and non-users that–while relatively small–may nonetheless be important for understanding how beliefs and knowledge relate to CE use. For example, whilst no statistically significant differences in knowledge were found between users and non-users, the 95% CIs for these effects imply that a 10% difference in knowledge is plausible for knowledge of safety (non-users better) or side effects (users better). We therefore recommend that future studies using similar

methods to ours take our sample size (~150) as a minimum target. Relatedly, we note here that the one analysis where we did not achieve 80% power to detect a large effect was the knowledge scale with the lowest degree of precision (5 questions, generating 6 levels of performance). Having fewer questions for this measure than the others may have reduced the signal-to-noise ratio associated with this measure, and reduced power accordingly. Therefore, future studies should–if feasible–aim for a more fine-grained measure of knowledge. Fourthly, whilst this study considered the types of sources users and non-users consulted for safety information, we did not investigate this further in terms of specific sources or the types of information they were extracting from the sources. This should be considered in future research. Finally, the current study focused on risks only and did not ask participants about their perceived benefits of using CE. Whilst this approach is supported in both theoretical models and applied health intervention [29, 53], future research could be expanded to consider benefits.

## Conclusions

A growing body of evidence has shown that users of CE believe the drugs to be safer than non-users [4, 12, 20] and that users more strongly believe that they know enough about the drugs to use them safely [12]. In the present study we have shown that the two groups consult similar sources of information when considering the safety of CE, indicating that differences in beliefs about safety are unlikely to be fully explained by use of different sources of information. Furthermore, both groups held comparable and relatively poor knowledge of the safety of the most reported CE, modafinil. Whilst, for non-users there were correlations between safety beliefs and safety knowledge, no significant relationships existed for users. This indicates that greater belief that CE are safe to use and that they have sufficient safety knowledge to make judgements in users is not based on more accurate safety knowledge. Given the lack of differences between sources of information and accuracy of knowledge, future research should consider the processes that mediate the relationship between evidence and beliefs, for example, examining the role of motivated reasoning in CE beliefs.

## Supporting information

**S1 File. Survey questions.**
(DOCX)

**S2 File. Understanding the relationship between safety beliefs and knowledge for cognitive enhancers in UK university students.**
(DOCX)

## Author Contributions

**Conceptualization:** Tim Rakow, Benjamin Gardner, Eleanor J. Dommett.

**Formal analysis:** Ngoc Trai Nguyen, Tim Rakow.

**Methodology:** Benjamin Gardner, Eleanor J. Dommett.

**Project administration:** Eleanor J. Dommett.

**Writing – original draft:** Eleanor J. Dommett.

**Writing – review & editing:** Tim Rakow, Benjamin Gardner, Eleanor J. Dommett.

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
