## [Decision Letter · Decision Letter 0]

6 May 2020

PONE-D-20-07694

Understanding the relationship between safety beliefs and knowledge for cognitive enhancers in UK university students

PLOS ONE

Dear Dr Dommett,

Thank you for submitting your manuscript to PLOS ONE. After careful consideration, we feel that it has merit but does not fully meet PLOS ONE’s publication criteria as it currently stands. Therefore, we invite you to submit a revised version of the manuscript that addresses the points raised during the review process.

It was very difficult to find reviewers willing to assess the manuscript. I was able, however, to collect feedback from a reviewer who provided what I consider as useful feedback to revise your manuscript. Please, see the comments at the bottom of this letter. Because this can be considered as a major review, please notice that a resubmission will require another round of reviews involving additional reviewers, and that the final outcome of this process is uncertain at this point.

We would appreciate receiving your revised manuscript by Jun 20 2020 11:59PM. To enhance the reproducibility of your results, we recommend that if applicable you deposit your laboratory protocols in protocols.io, where a protocol can be assigned its own identifier (DOI) such that it can be cited independently in the future. For instructions see: http://journals.plos.org/plosone/s/submission-guidelines#loc-laboratory-protocols

We look forward to receiving your revised manuscript.

Kind regards,

Angel Blanch, Ph.D.

Academic Editor

PLOS ONE

Journal Requirements:

Reviewers' comments:

Reviewer's Responses to Questions

**Comments to the Author**

1. Is the manuscript technically sound, and do the data support the conclusions?

Reviewer #1: No

2. Has the statistical analysis been performed appropriately and rigorously? 

Reviewer #1: Yes

3. Have the authors made all data underlying the findings in their manuscript fully available?

Reviewer #1: No

4. Is the manuscript presented in an intelligible fashion and written in standard English?

Reviewer #1: Yes

5. Review Comments to the Author

Reviewer #1: This study examined how beliefs about the safety of cognitive enhancers (CEs), knowledge about their safety, and sources of relevant knowledge differed between users and non-users of CEs. One hundred forty-eight university students completed an online questionnaire. The results showed that 21% of the students had used CEs, and they highly evaluated the safety of CEs compared to non-users. On the other hand, sources of information on CE safety did not differ for users and non-users, suggesting that the higher safety levels perceived by users was not because information sources were different from those of non-users. There was also no significant difference between the groups regarding knowledge of CE safety. In addition, there was no significant correlation between safety beliefs and safety knowledge among CE users.

This study addresses an interesting issue, and the finding that the basis for the safety beliefs of CE users is weak has considerable social significance. However, there are concerns, mainly about the validity of the results. The authors should consider the following three issues and make the necessary corrections:

1) Since the findings of this study are based on null results, the interpretation of the results should be performed carefully. In this study, the sample of CE users was only 30 of 148 students who cooperated by completing the online survey. What is the statistical power in this case? One of the important findings of this study is that the number of sources and the degree of utilization of each source regarding CE safety knowledge did not differ between CE users and non-users. However, in some analyses, the p-value was shown to be close to a significant value. Therefore, there is the possibility that the p-value did not become significant because of insufficient statistical power. Thus, the evidence is too weak to conclude that there was no difference in the sources of safety knowledge for CE users and non-users. The result that there was no difference in safety knowledge between the groups should be also considered carefully. The authors should show the results of the power analysis to indicate that the size of the sample was sufficient for detecting a significant difference. Otherwise, data should be collected from more participants (especially CE users) based on sample size determination. The author stated in line 377 that “the overall sample size was sufficient for statistical analysis,” but the basis for this reasoning is unclear.

2) Although the authors aimed to compare CE users and non-users, there is no table or graph showing the data for each. Although the Results section explains whether the differences were significant, it is important to show which group scored higher. The authors should present data by group.

3) Human behavior is determined by considering not only risks but also benefits. However, in this study, there were three questions focused only on the risks associated with modafinil; they were included to examine participants’ knowledge regarding the safety of CEs. This approach was also used to examine beliefs about CEs. The authors should explain why they focused only on risks and not benefits.

6. PLOS authors have the option to publish the peer review history of their article (what does this mean?). If published, this will include your full peer review and any attached files.

Reviewer #1: No

---

## [Author Response · Author response to Decision Letter 0]

10 Jun 2020

The information below is a copy of the Response to Reviewers document.

Response to Reviewers

Reviewer #1: This study examined how beliefs about the safety of cognitive enhancers (CEs), knowledge about their safety, and sources of relevant knowledge differed between users and non-users of CEs. One hundred forty-eight university students completed an online questionnaire. The results showed that 21% of the students had used CEs, and they highly evaluated the safety of CEs compared to non-users. On the other hand, sources of information on CE safety did not differ for users and non-users, suggesting that the higher safety levels perceived by users was not because information sources were different from those of non-users. There was also no significant difference between the groups regarding knowledge of CE safety. In addition, there was no significant correlation between safety beliefs and safety knowledge among CE users. This study addresses an interesting issue, and the finding that the basis for the safety beliefs of CE users is weak has considerable social significance. However, there are concerns, mainly about the validity of the results. The authors should consider the following three issues and make the necessary corrections:

1) Since the findings of this study are based on null results, the interpretation of the results should be performed carefully. In this study, the sample of CE users was only 30 of 148 students who cooperated by completing the online survey. What is the statistical power in this case? One of the important findings of this study is that the number of sources and the degree of utilization of each source regarding CE safety knowledge did not differ between CE users and non-users. However, in some analyses, the p-value was shown to be close to a significant value. Therefore, there is the possibility that the p-value did not become significant because of insufficient statistical power. Thus, the evidence is too weak to conclude that there was no difference in the sources of safety knowledge for CE users and non-users. The result that there was no difference in safety knowledge between the groups should be also considered carefully. The authors should show the results of the power analysis to indicate that the size of the sample was sufficient for detecting a significant difference. Otherwise, data should be collected from more participants (especially CE users) based on sample size determination. The author stated in line 377 that “the overall sample size was sufficient for statistical analysis,” but the basis for this reasoning is unclear.

Thank you for this comment, and for this reminder to report and discuss the precision of our estimation in the manuscript. In keeping with current recommendations for this, we now include confidence intervals for each of the mean difference between users and non-users that we report (Cumming, 2014). This means that readers can see what the likely limits are for the effects that we report, and can judge whether any effects that significance tests may have failed to detect might be sufficiently large to warrant further investigation. You will see from the confidence intervals that we report that our sample size of 148 allowed for reasonable precision in estimation, such that it is highly unlikely that we have failed to detect large effects. Additionally, we now acknowledge and discuss that we have limited power to detect small effects. In keeping with the reviewer’s comment above, we do that in relation to the non-significant effect for which the p-value came closest to .05. We also discuss statistical power, more generally, in our limitations section and include recommendations for other researchers. 

2) Although the authors aimed to compare CE users and non-users, there is no table or graph showing the data for each. Although the Results section explains whether the differences were significant, it is important to show which group scored higher. The authors should present data by group.

We have now replaced the original Figure 1 with a figure that divides the data by group. This provides by group data for Aim 1. We have also added an additional Figure (Figure 2) to provide by group data relating to Aim 2. Aim 3 data is provided overall in a table and correlations by group are in the text. Therefore, by group data is now presented for all aims. 

3) Human behavior is determined by considering not only risks but also benefits. However, in this study, there were three questions focused only on the risks associated with modafinil; they were included to examine participants’ knowledge regarding the safety of CEs. This approach was also used to examine beliefs about CEs. The authors should explain why they focused only on risks and not benefits.

We have now included a paragraph within the introduction explaining our rationale for this approach. To briefly summarise this here we have explained that when studying drug safety, including safety of CE, the focus is typically on risk rather than benefits, and that the benefits are not very well understood, even in key populations. Additionally, theoretical models, supported by substantial experimental data, indicate that risks or potential losses are more important than benefits or gains, and health behaviour interventions that modulate risk perception, rather than benefits, are more likely to be effective. However, we have also included in the discussion that this restriction to risks is a limitation of the current study.

Additional Requirements

1. Please ensure that your manuscript meets PLOS ONE's style requirements, including those for file naming. The PLOS ONE style templates can be found at:

We have made amendments to the manuscript to bring this in line with the requirements.

We have now provided a full copy of the survey used as supporting information and made this clear from the text.

This is still the case. Our institutional repository will only include published data. Therefore, on acceptance we will publish the data with them and provide the DOI.

Reference

Cumming, G. (2014). The new statistics: Why and how. Psychological Science, 25(1), 7–29. https://doi.org/10.1177/0956797613504966

---

## [Decision Letter · Decision Letter 1]

20 Jul 2020

PONE-D-20-07694R1

Understanding the relationship between safety beliefs and knowledge for cognitive enhancers in UK university students

PLOS ONE

Dear Dr. Dommett,

Thank you for submitting your manuscript to PLOS ONE. After careful consideration, we feel that it has merit but does not fully meet PLOS ONE’s publication criteria as it currently stands. Therefore, we invite you to submit a revised version of the manuscript that addresses the points raised during the review process.

This version of the manuscript has been re-evaluated by the same Reviewer who did the initial review. As you will see in the comments appended below, the Reviewer was unconvinced that the initial raised concerns were properly addressed and is recommending to reject the manuscript at this point. After my own reading of the manuscript, however, I do think that these concerns could be addressed in another reviewed version of the study.

We look forward to receiving your revised manuscript.

Kind regards,

Angel Blanch, Ph.D.

Academic Editor

PLOS ONE

Reviewers' comments:

Reviewer's Responses to Questions

**Comments to the Author**

1. If the authors have adequately addressed your comments raised in a previous round of review and you feel that this manuscript is now acceptable for publication, you may indicate that here to bypass the “Comments to the Author” section, enter your conflict of interest statement in the “Confidential to Editor” section, and submit your "Accept" recommendation.

Reviewer #1: (No Response)

2. Is the manuscript technically sound, and do the data support the conclusions?

Reviewer #1: No

3. Has the statistical analysis been performed appropriately and rigorously? 

Reviewer #1: No

4. Have the authors made all data underlying the findings in their manuscript fully available?

Reviewer #1: Yes

5. Is the manuscript presented in an intelligible fashion and written in standard English?

Reviewer #1: Yes

6. Review Comments to the Author

Reviewer #1: This study investigated the relationship between the safety beliefs on the use of cognitive enhancers (CE), the sources of knowledge relating to their safety, and the existence of actual knowledge among its users as well as non-users. Of the three points to which I had drawn attention to, in the first review; the second and third were appropriately addressed, but the first was answered only partially, leading to an inadequacy in this paper.

In the first review, I had recommended to the authors to either show that the results of the power analysis based on the present sample size were sufficient for detecting a significant difference or to collect additional data, particularly on CE users, based on sample size determination. However, they neither implemented my recommendations nor explained their rationale for not doing so; but instead showed a 95% confidence interval (CI) for the mean differences. While I agree that it is beneficial to report the 95% CI, this revision is not a solution to the issue that I pointed out, since CI is an index of the accuracy of the differences of mean values rather than an index of statistical power.

One of the main objectives of this study was to clarify whether there were differences in the sources of knowledge on the safety of CE between its users and non-users. However, the results showed no significant differences between the two groups relating to the number of reported sources. Regarding this result, the authors argued that even if a difference actually existed in the number of sources between the CE users and non-users, it could have been just one or two, based on the 95% CI of the mean difference (lines 328–333), which they did not consider as a large difference. However, this interpretation is arbitrary because there is no basis for their considering the difference as not being large.

More importantly, the revised version of Figure 1 shows that compared to CE non-users, its users were more likely to use the experiences of peers and websites as sources of information on CE safety. Also, the opposite trends can be seen for social media, NICE guidelines, and scientific research. The p-value of these differences was around 0.15 in several cases, which suggests some meaningful differences, although it is a small effect. While this is an important trend relating to the objective of this study; the reliability of this difference is unclear because its statistical power was weak. Since the authors were unable to provide a clear answer relating to the study’s main objective, I feel that, as of now, this study is not publication-ready.

To overcome the above-mentioned ambiguity, the authors should carry out the research and resubmit it, after increasing the number of participants.

7. PLOS authors have the option to publish the peer review history of their article (what does this mean?). If published, this will include your full peer review and any attached files.

Reviewer #1: No

---

## [Author Response · Author response to Decision Letter 1]

16 Oct 2020

Please see response to reviewer document to see this in full (i.e. with colour coding referred to):

Response to Reviewers

Reviewer #1: This study investigated the relationship between the safety beliefs on the use of cognitive enhancers (CE), the sources of knowledge relating to their safety, and the existence of actual knowledge among its users as well as non-users. Of the three points to which I had drawn attention to, in the first review; the second and third were appropriately addressed, but the first was answered only partially, leading to an inadequacy in this paper. In the first review, I had recommended to the authors to either show that the results of the power analysis based on the present sample size were sufficient for detecting a significant difference or to collect additional data, particularly on CE users, based on sample size determination. However, they neither implemented my recommendations nor explained their rationale for not doing so; but instead showed a 95% confidence interval (CI) for the mean differences. While I agree that it is beneficial to report the 95% CI, this revision is not a solution to the issue that I pointed out, since CI is an index of the accuracy of the differences of mean values rather than an index of statistical power.

One of the main objectives of this study was to clarify whether there were differences in the sources of knowledge on the safety of CE between its users and non-users. However, the results showed no significant differences between the two groups relating to the number of reported sources. Regarding this result, the authors argued that even if a difference actually existed in the number of sources between the CE users and non-users, it could have been just one or two, based on the 95% CI of the mean difference (lines 328–333), which they did not consider as a large difference. However, this interpretation is arbitrary because there is no basis for their considering the difference as not being large.

More importantly, the revised version of Figure 1 shows that compared to CE non-users, its users were more likely to use the experiences of peers and websites as sources of information on CE safety. Also, the opposite trends can be seen for social media, NICE guidelines, and scientific research. The p-value of these differences was around 0.15 in several cases, which suggests some meaningful differences, although it is a small effect. While this is an important trend relating to the objective of this study; the reliability of this difference is unclear because its statistical power was weak. Since the authors were unable to provide a clear answer relating to the study’s main objective, I feel that, as of now, this study is not publication-ready. To overcome the above-mentioned ambiguity, the authors should carry out the research and resubmit it, after increasing the number of participants.

Response: We are pleased that the reviewer noted we had addressed two of their concerns with this revision. For the concern about power we have made some further amendments to the manuscript, but we respectively disagree with the reviewer about this issue and explain this fully below. 

In keeping with current recommendations for best practice (e.g., Cumming, 2014), we included confidence intervals for effects wherever it was straightforward to do so. There is a direct link between confidence intervals and power calculations because both provide information about possible effects in the population, and both rely on the standard error: the width of a confidence interval is determined by the standard error, as is the power to detect a population effect of a given size. Therefore, when – for example – an analysis has high power to detect a small effect, the confidence interval for that effect will be very narrow; while when the power to detect a moderate or large effect is poor the confidence interval will be very wide. To illustrate, suppose we have a confidence interval for a mean difference of width 9.99 units, and the true effect is 10.00 units. By definition of the CI, we expect 95/100 studies to have confidence intervals that include the true effect of 10.00. it follows from the interval width that those 95/100 studies will all return significant results because the confidence interval must exclude zero (an interval that is 9.99 wide cannot include both 0 and 10.00). Therefore, the probability of obtaining a significant effect exceeds 0.95, and therefore it follows from the definition of statistical power that the power to detect this effect exceeds .95., We regret not making that link explicit in our previous version, but have done so in our current revision, as follows (pages 13 and 14, red font represents the main revision to this portion):

While the mean number of sources used was higher for users (M = 3.43, SD = 1.14) than non-users (M = 2.97, SD = 1.21), this difference was not statistically significant (t(145) = 1.91, p=0.058, 95% CI -0.95, 0.01) although the effect was small-to-medium in size (d=0.40). Thus, we did not detect a difference between users and non-users in the mean number of information sources used. An implication of the width of the 95% CI for this difference (width just below 1) is that our study had excellent statistical power (>.95) to detect a difference of 1 additional data source.

For none of these measures did the mean number correct differ significantly between users and non-users (Fig 2): Not Safe, t(146)=1.35, p=0.179, 95% CI –0.69, 0.13; Monitor, t(146)=0.72, p=0.471, 95% CI -0.72, 0.34; Side Effects, t(146)=1.38, p=0.171, 95% CI -0.30, 1.68. The confidence intervals for these mean differences are for scales with a maximum possible score of 5, 9 and 15, respectively; and therefore the 95% CIs span 19.8%, 11.8% and 13.2% of each scale-range, respectively. These spans therefore represent population effects for mean differences for which our sample had excellent statistical power (>.95) while for effects which might be considered small – differences in the region of 6% to 10% of the scale-range – our CIs imply modest statistical power (≈.50).

Notice that in the second extract we have expressed the potential effect in as a proportion of the scale range. In doing so, we align with the reviewer’s comment that any designation of an effect as ‘small’, ‘medium’ or ‘large’ is either conventional or arbitrary. This description of the potential effect is much more conducive to reader’s exercising their own judgment in these matters. Personally, we would regard difference of 15-20% of these scales as important, and differences of around 6-10% as modest. However, these way of presenting the information allows readers to exercise their own judgment.

We respectfully disagree with the reviewer’s view that p-values around 0.15 might be taken as evidence of an effect. We prefer to adopt a more conservative and conventional interpretation of p-values, which is that we simply should not speculate about the possible presence or direction of an effect in such circumstances. This is very much in keeping with the weight of reviewer opinion in our experience, wherein we have regularly seen authors chastised by reviewers for describing effects with p = .07 or p = .08 as ‘approaching significance’ or ‘marginally significant’. We may indeed have missed a small effect – but p ≈ .15 cannot be taken as evidence that we should be confident the direction of any such effect, should it exist. This conservatism is in line with current concerns about reproducibility and replication in psychological science where many effects that have been identified with p = .03 or .04 have subsequently been found to be non-replicable (e.g., Munafo, Nosek, Bishop et al., 2017). Perhaps also relevant to mention here, the (non-significant) mean difference of around one-half is entirely consistent with the significant difference that we find and report whereby approximately an additional 50% of CE users use personal information as a source of information.

Related to these issues, the reviewer argues that we have not met one of the main objectives of our study: “One of the main objectives of this study was to clarify whether there were differences in the sources of knowledge on the safety of CE between its users and non-users.” This does not quite match with how we expressed our objective concerning sources of knowledge. We said that we aimed to compare sources of information between users and non-users, and demonstrably we have done this. Additionally, the reviewer seems to interpret our discussion to mean that the true difference in the number of information sources could have been one or two. However, we said that it is unlikely to be as large as one or two. We thought we had been clear in what we had written, however, hopefully the additional comment in the Results about this CI (see above) provides additional clarity on this. 

Cumming, G. (2014). The new statistics: why and how. Psychological Science, 25(1), 7–29. https://doi.org/10.1177/0956797613504966

Munafo, M. R., Nosek, B. A., Bishop, D. V. M., et al., (2017). A manifesto for reproducible science. Nature Human Behaviour, 1:0021 DOI: 10.1038/s41562-016-0021

---

## [Decision Letter · Decision Letter 2]

30 Oct 2020

PONE-D-20-07694R2

Understanding the relationship between safety beliefs and knowledge for cognitive enhancers in UK university students

PLOS ONE

Dear Dr. Dommett,

Thank you for submitting your manuscript to PLOS ONE. After careful consideration, we feel that it has merit but does not fully meet PLOS ONE’s publication criteria as it currently stands. Therefore, we invite you to submit a revised version of the manuscript that addresses the points raised during the review process.

This version of the manuscript has been evaluated by a fresh new Reviewer (#2) who provided some further suggestions. Please, see the specific comments at the bottom of this letter. As you will see, there were some major concerns with the statistical approach to analyze the data. These concerns should be addressed in another reviewed version of your study.

We look forward to receiving your revised manuscript.

Kind regards,

Angel Blanch, Ph.D.

Academic Editor

PLOS ONE

Reviewers' comments:

Reviewer's Responses to Questions

**Comments to the Author**

1. If the authors have adequately addressed your comments raised in a previous round of review and you feel that this manuscript is now acceptable for publication, you may indicate that here to bypass the “Comments to the Author” section, enter your conflict of interest statement in the “Confidential to Editor” section, and submit your "Accept" recommendation.

Reviewer #2: (No Response)

2. Is the manuscript technically sound, and do the data support the conclusions?

Reviewer #2: Yes

3. Has the statistical analysis been performed appropriately and rigorously? 

Reviewer #2: No

4. Have the authors made all data underlying the findings in their manuscript fully available?

Reviewer #2: No

5. Is the manuscript presented in an intelligible fashion and written in standard English?

Reviewer #2: Yes

6. Review Comments to the Author

Reviewer #2: The present questionnaire-based study compared the users and non-users of cognitive enhancers (CE) in terms of their knowledge, accuracy, and safety beliefs about the CE use.

The introduction is written in a comprehensive manner, citing relevant literature, the methods are well-described. The issues that threaten the quality of the study can be found only in the applied statistical procedures.

Major issues:

The analyzed sample consisted of 148 participants out of which 21% identified themselves as CE users. Although this data may be representative of the studied population, there is a disproportion in the size of the compared groups. This raises concerns about the homogeneity of variance of the compared groups that is an important assumption for the used ANOVAs.

The manuscript should therefore explicitly mention how they dealt with this limitation and whether and why are the used statistical methods appropriate.

The different group sizes are also related to another, previously discussed, issue: the statistical power. When I used the widely recommended (e.g. Cumming, 2014) G*Power software (available at tiny.cc/gpower3) to calculate the power of the study given the provided data, the power was much lower than the one reported in the manuscript (0.95). Please note that I do not recommend post-hoc power calculation. However, it would considerably increase the quality of the manuscript, if the authors included information about the probability of finding predicted difference, given the study parameters, which is not be based only on the confidence intervals.

Minor issues:

There are a few cases of misplaced punctuation and inconsistency in citation style.

7. PLOS authors have the option to publish the peer review history of their article (what does this mean?). If published, this will include your full peer review and any attached files.

Reviewer #2: **Yes: **Hana H. Kutlikova

---

## [Author Response · Author response to Decision Letter 2]

14 Dec 2020

Note that this is clearer in the attachment because of the statistics but we have copied in here as well.

Thank you for the review of the previous version of our manuscript. We have now fully revised our manuscript in line with the reviewer comments. In tandem with these changes to the main manuscript, we now provide additional details of power calculations and statistical analyses in Supporting Information (S2). Please see below for our response to each reviewer point.

1. The analyzed sample consisted of 148 participants out of which 21% identified themselves as CE users. Although this data may be representative of the studied population, there is a disproportion in the size of the compared groups. This raises concerns about the homogeneity of variance of the compared groups that is an important assumption for the used ANOVAs. The manuscript should therefore explicitly mention how they dealt with this limitation and whether and why are the used statistical methods appropriate.

Thank you for this comment. Our Data Analysis section now confirm explicitly that equal variances were assumed for these analyses and point readers to our new Supporting Information (S2) that report a robustness checks of each between-subjects test of mean differences that do not assume equal variances:

For these and other between-subjects tests of mean difference we assume homogeneity of variance in the population. Analyses reported in the Supporting Information (S2) show that no conclusions change if this assumption is not made.

The Supporting Information (S2) explain that the assumption of equal variances in the population is reasonable because we had no a priori reason to depart from this assumption, and because the observed differences between groups in sample variance were small. Nonetheless, the point is well made that any issues arising from heterogeneity in variances are exacerbated when the sample sizes are unequal. Therefore, we think it prudent to include analyses in our Supporting Information (S2) which demonstrate that analyses that assume unequal variances yield the same conclusions as analysis that assume equal variances.

2. The different group sizes are also related to another, previously discussed, issue: the statistical power. When I used the widely recommended (e.g. Cumming, 2014) G*Power software (available at tiny.cc/gpower3) to calculate the power of the study given the provided data, the power was much lower than the one reported in the manuscript (0.95). Please note that I do not recommend post-hoc power calculation. However, it would considerably increase the quality of the manuscript, if the authors included information about the probability of finding predicted difference, given the study parameters, which is not be based only on the confidence intervals.

Thank you for suggesting that we use G*Power software. This has allowed us to provide details of statistical power in a format that will be more familiar to readers. We agree with the reviewer that there is limited value to a post hoc power calculation (e.g., based on the observed effect size). We have therefore based our power calculations on the power to detect effects in the population that we would not want to miss. These were specified in terms of unstandardised mean differences, because these have a meaningful and concrete interpretation in the context of our study (see Baguley, 2009). This required that we use sample data to estimate the standard deviation of scores in the population (as is almost always the case in the behavioural sciences). These calculations took account of the unequal sample sizes. Our Supporting Information (S2) describe this process in detail.

We have replaced the previous description of power in the Results with descriptions based the power calculations in G*Power. These confirm that most analyses had very good statistical power to detect the effects that we specified for the calculations.

On page 13, we now write:

Using G*Power software, we determined that our sample size was sufficient for 0.95 power to detect a mean difference of one additional data source in the population using a two-tailed test (see Supporting Information (S2) for full details). Therefore, if a difference of that scale or larger were to exist in the population, it is highly unlikely that we would fail to detect a difference.

And on page 15:

We used G*Power software to determine the sample sizes required for 0.8 power to detect an absolute mean difference of 10% of the scale-range for each knowledge scores. Our sample size exceeds the sample size required for this level of statistical power for the monitor and not safe scores, though an additional 56 participants would be required to achieve this high degree of power for the not safe scale (see Supporting Information (S2)).As indicated in these additions, fuller details are given in the Supporting Information (S2).

We have also added this short comment to the Limitations sub-section (page 21) because our re-calculation of power highlighted a potential additional route to improve measurement precision and statistical power.

Relatedly, we note here that the one analysis where we did not achieve 80% power to detect a large effect was the knowledge scale with the lowest degree of precision (5 questions, generating 6 levels of performance). Having fewer questions for this measure than the others may have reduced the signal-to-noise ratio associated with this measure, and reduced power accordingly. Therefore, future studies should – if feasible – aim for a more fine-grained measure of knowledge.

For transparency, we have appended the protocols for each power analysis at the end of this letter.

3. There are a few cases of misplaced punctuation and inconsistency in citation style.

We have checked the manuscript for punctuation and style. Additionally, we have corrected one rounding error in the reporting of an effect size (d = 0.40 corrected to d = 0.39 on page 13).

Kind regards,

Eleanor Dommett and colleagues

Reference

Baguley, T., 2009. Standardized or simple effect size: what should be reported? British Journal of Psychology, 100(3), 603-617.

Protocols for power calculations using G*Power 

1. Sample size required for 0.95 power to detect a mean difference of 1 additional information sources (estimated standardised effect of d = 0.834) 

[1] -- Monday, December 14, 2020 -- 12:52:07

t tests - Means: Difference between two independent means (two groups)

Analysis: A priori: Compute required sample size 

Input: Tail(s) = Two

 Effect size d = 0.834

 α err prob = 0.05

 Power (1-β err prob) = 0.95

 Allocation ratio N2/N1 = 3.9

Output: Noncentrality parameter δ = 3.6466543

 Critical t = 1.9806260

 Df = 116

 Sample size group 1 = 24

 Sample size group 2 = 94

 Total sample size = 118

 Actual power = 0.9511731

2. Sample size required for 0.8 power to detect a mean difference in knowledge score (not safe) of 10% of the response scale (mean difference of 0.5, estimated standardised effect of d = 0.491).

[2] -- Monday, December 14, 2020 -- 13:03:29

t tests - Means: Difference between two independent means (two groups)

Analysis: A priori: Compute required sample size 

Input: Tail(s) = Two

 Effect size d = 0.491

 α err prob = 0.05

 Power (1-β err prob) = 0.80

 Allocation ratio N2/N1 = 3.933

Output: Noncentrality parameter δ = 2.8102965

 Critical t = 1.9717774

 Df = 202

 Sample size group 1 = 41

 Sample size group 2 = 163

 Total sample size = 204

 Actual power = 0.7986900

3. Sample size required for 0.8 power to detect a mean difference in knowledge score of 10% of the response scale (mean difference of 0.9, estimated standardised effect of d = 0.686).

[3] -- Monday, December 14, 2020 -- 13:05:00

t tests - Means: Difference between two independent means (two groups)

Analysis: A priori: Compute required sample size 

Input: Tail(s) = Two

 Effect size d = 0.686

 α err prob = 0.05

 Power (1-β err prob) = 0.8

 Allocation ratio N2/N1 = 3.933

Output: Noncentrality parameter δ = 2.8150771

 Critical t = 1.9830375

 Df = 104

 Sample size group 1 = 21

 Sample size group 2 = 85

 Total sample size = 106

 Actual power = 0.7964569

4. Sample size required for 0.8 power to detect a mean difference in knowledge score of 10% of the response scale (mean difference of 1.5, estimated standardised effect of d = 0.686).

[4] -- Monday, December 14, 2020 -- 13:06:57

t tests - Means: Difference between two independent means (two groups)

Analysis: A priori: Compute required sample size 

Input: Tail(s) = Two

 Effect size d = 0.612

 α err prob = 0.05

 Power (1-β err prob) = 0.8

 Allocation ratio N2/N1 = 3.933

Output: Noncentrality parameter δ = 2.8362270

 Critical t = 1.9783804

 Df = 130

 Sample size group 1 = 27

 Sample size group 2 = 105

 Total sample size = 132

 Actual power = 0.8037949

---

## [Editor Report · Decision Letter 3]

18 Dec 2020

Understanding the relationship between safety beliefs and knowledge for cognitive enhancers in UK university students

PONE-D-20-07694R3

Dear Dr. Dommett,

We’re pleased to inform you that your manuscript has been judged scientifically suitable for publication and will be formally accepted for publication once it meets all outstanding technical requirements.

Kind regards,

Angel Blanch, Ph.D.

Academic Editor

PLOS ONE
---

## [Editor Report · Acceptance letter]

20 Jan 2021

PONE-D-20-07694R3 

Understanding the relationship between safety beliefs and knowledge for cognitive enhancers in UK university students 

Dear Dr. Dommett:

I'm pleased to inform you that your manuscript has been deemed suitable for publication in PLOS ONE. Congratulations! Your manuscript is now with our production department. 

Kind regards, 

on behalf of

Dr. Angel Blanch 

Academic Editor

PLOS ONE